# Adaptive Multi-step Refinement Network for Robust Point Cloud Registration

**Zhi Chen**[*]                                                                 *z_chen@hust.edu.cn*
*HUST*

**Yufan Ren**[*]                                                                *yufan.ren@epfl.ch*
*IVRL, EPFL*

**Tong Zhang**                                                              *tong.zhang@epfl.ch*
*IVRL, EPFL*

**Zheng Dang**                                                             *zheng.dang@epfl.ch*
*CVLab, EPFL*

**Wenbing Tao**                                                        *wenbingtao@hust.edu.cn*
*HUST*

**Sabine Süsstrunk**                                                  *sabine.susstrunk@epfl.ch*
*IVRL, EPFL*

**Mathieu Salzmann**                                              *mathieu.salzmann@epfl.ch*
*CVLab, EPFL*

**Reviewed on OpenReview:** *https://openreview.net/forum?id=M3SkSMfWcP*

## Abstract

Point Cloud Registration (PCR) estimates the relative rigid transformation between two point clouds of the same scene. Despite significant progress with learning-based approaches, existing methods still face challenges when the overlapping region between the two point clouds is small. In this paper, we propose an adaptive multi-step refinement network that refines the registration quality at each step by leveraging the information from the preceding step. To achieve this, we introduce a training procedure and a refinement network. Firstly, to adapt the network to the current step, we utilize a generalized one-way attention mechanism, which prioritizes the last step's estimated overlapping region, and we condition the network on step indices. Secondly, instead of training the network to map either random transformations or a fixed pre-trained model's estimations to the ground truth, we train it on transformations with varying registration qualities, ranging from accurate to inaccurate, thereby enhancing the network's adaptiveness and robustness. Despite its conceptual simplicity, our method achieves state-of-the-art performance on both the 3DMatch/3DLoMatch and KITTI benchmarks. Notably, on 3DLoMatch, our method reaches **80.4%** recall rate, with an absolute improvement of **1.2%**. Our code is available at https://github.com/ZhiChen902/AMR.

## 1 Introduction

Point Cloud Registration (PCR) estimates the optimal rigid transformation between two point clouds. The classical Iterative Closest Point (ICP) algorithm (Besl & McKay, 1992) pairs points in two overlapping point

---

[*]Equal contribution

clouds and minimizes the pairwise Euclidean distances. ICP employs an iterative optimization process and, due to its non-convex nature, requires precise initialization. Some ICP variants (Yang et al., 2015b; Campbell & Petersson, 2016) introduce global optimizations to improve robustness, at the cost of low efficiency (Fu et al., 2023).

Recently, learning-based methods, particularly feature-matching techniques, have become predominant in PCR (Choy et al., 2019; Huang et al., 2021; Qin et al., 2022; Yang et al., 2022; Yu et al., 2021; 2023b; Ao et al., 2023). These methods employ neural networks to extract features of key points (down-sampled point cloud), establish point correspondences (Yu et al., 2021; Qin et al., 2023; Yang et al., 2022), and utilize robust estimators to determine the relative transformation (Choy et al., 2020; Bai et al., 2021; Fischler & Bolles, 1981; Chen et al., 2022a). While these learning-based methods have shown impressive performance, they encounter challenges in low overlapping scenarios, typically $< 30\%$. The primary issue is during the feature extraction phase, similar structures and shapes may produce ambiguous key point features. Consequently, feature-based methods may erroneously match key points that do not belong to the overlapping region, thereby introducing inaccuracies, as is depicted in Fig. 1 (a) [1]. The source point cloud, colored in yellow, represents the left side of a kitchen scene, while the target point cloud, in blue, represents the right side of the room. The overlapping region between these two point clouds is small, as can be seen on the right-most of Fig. 1 (c), where the two point clouds are almost perfectly aligned. As shown by the red lines in Fig. 1 (a), the seminal work GeoTransformer(Qin et al., 2022) erroneously matches one point in the source point cloud to points outside the overlapping region.

The recent work PEAL (Yu et al., 2023b) introduces a post-processing method to refine the registration results. It utilizes the overlap estimation of a pre-trained PCR network, i.e., GeoTransformer(Qin et al., 2022), as auxiliary information (also referred to as prior), and incorporates this information in the refinement network to improve the estimate. By employing the same network repeatedly, PEAL establishes a multi-step approach (see Fig. 1 (b)) and demonstrates enhanced performance. However, despite being utilized in multiple steps, the refinement network is only trained for a fixed mapping from prior to ground truth, not considering the fact that the priors become increasingly accurate throughout the refinement steps.

In this work, our intuition is to enhance multi-step registration by making the refinement process adaptive. Our first contribution is a training procedure that, rather than mapping a random transformation or a pre-trained model's estimate (i.e., GeoTransformer) to the ground truth, trains the network with priors of varying accuracy levels, from accurate to poor alignments. We design a degradation function that interpolates between the prior and ground truth, enabling a smooth transition from accurate to inaccurate priors. Adjusting the interpolation ratio lets us manipulate the quality of the priors. Furthermore, we propose to explicitly condition the network on the current refinement step index, with each step's network trained separately, reflecting the accuracy of the prior. Moreover, we propose a generalized version of one-way attention focusing on the overlapping regions in both the source and target point clouds. Considering the non-linearity of the transformation space, we define the degradation function in the spherical linear space (Yew & Lee, 2022).

Our extensive experiments demonstrate our method's effectiveness; we achieve state-of-the-art recall rates on the 3DMatch/3DLoMatch and KITTI benchmarks. Notably, our method yields **80.4%** recall rate on the 3DLoMatch benchmark, with an absolute improvement of **1.2%** compared with the previous state of the art. In summary, the effectiveness of our work stems from the following contributions:

- We introduce a novel adaptive multi-step refinement method tailored for the low-overlap challenge in PCR, achieving state-of-the-art registration recall rates on the 3DMatch/3DLoMatch and KITTI benchmarks.

- We propose a training procedure that makes the network adaptive to the improving prior quality during refinement and consequently improving registration.

- We propose a generalized one-way attention mechanism that focuses on the pre-aligned overlapping regions of the two point clouds, facilitating improved feature learning.

---

[1]We manually adjust the position and rotation of the two point clouds for better visualization of matches, same for Fig. 1 (b).

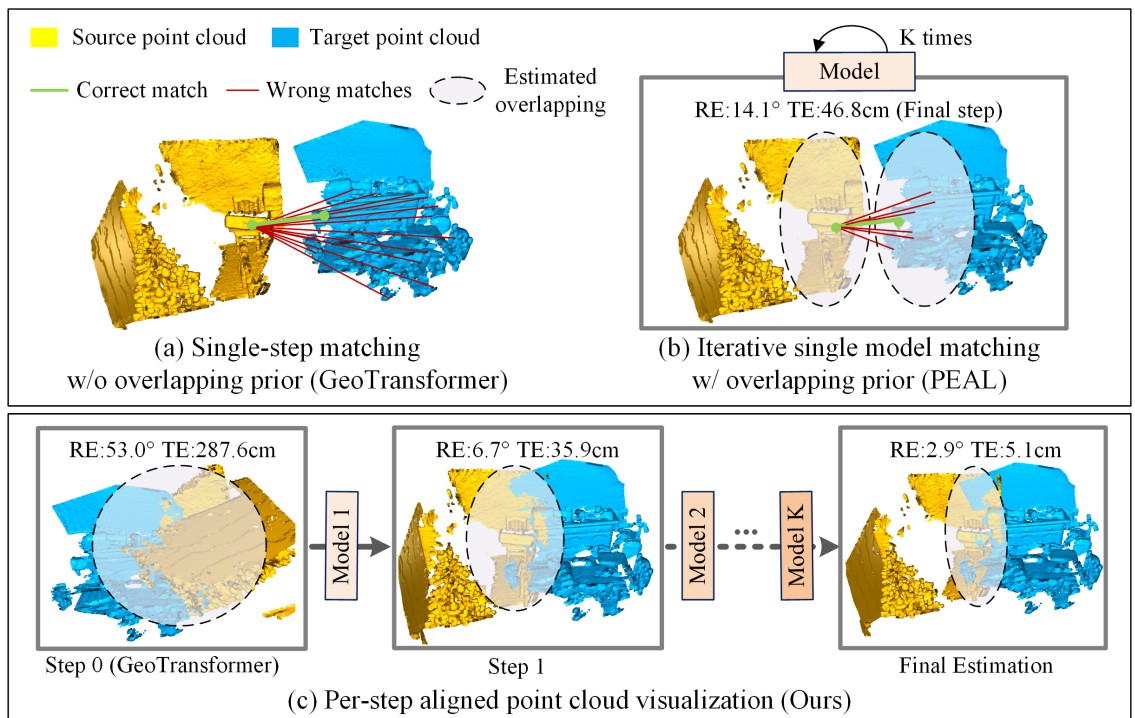

Figure 1: Illustration of the proposed framework. (a) Matching visualization. Similar structures and shapes result in similar features, leading to the erroneous matches by the GeoTransformer, where one key point in the source point is matched to key points in the target point cloud outside the overlapping region, indicated by red lines. The ground truth match is shown as a green line. (b) Utilizing rough overlapping information, indicated as the shaded regions, PEAL successfully filters out many incorrect matches outside the overlap area, visualized similarly to (a). (c) Per-step aligned point cloud visualization. Our adaptive multi-step refinement approach employs multiple models, each progressively trained with more accurate priors as the model index increases. By adapting to the progressively increasing accuracy of priors during the refinement steps, our method surpasses PEAL in performance. RE and TE stand for rotation error (↓) and translation error (↓) (Best viewed on a screen when zoomed in).

## 2  Related Work

**Traditional point cloud registration (PCR) methods** estimate the rigid transformation (rotation and translation) between two point clouds using geometric or appearance information. The most widely used PCR methods are Iterative Closest Point (ICP) (Besl & McKay, 1992) and its variants (Chen & Medioni, 1992; Segal et al., 2009). Despite their widespread application in various real-world scenarios due to their straightforward formulation, ICP algorithms are sensitive to initial conditions and are prone to convergence at local minimum. To address this, some approaches, such as Go-ICP (Yang et al., 2015b) and GOGMA (Campbell & Petersson, 2016), integrate Branch-and-Bound optimization into ICP. However, the global optimization process inherent to these methods significantly increases computational time, thus limiting their practicality.

An alternative methodology in PCR is the feature-matching-based approach. The methods in this category form correspondences by creating and matching point feature descriptors, and then use robust estimators to remove outliers and recover the relative pose. Based on how feature descriptors are obtained, traditional feature matching can be categorized into Local Reference Frame (LRF) based and non-LRF-based descriptors. LRF-based descriptors initially typically begin with covariance analysis (Novatnack & Nishino, 2008; Tombari et al., 2010) to create a local coordinate system, which then serves as the reference for generating feature descriptors. Techniques for deriving descriptors from LRF include coordinate plane projection (Zaharescu et al., 2009) and rotational projection statistics (Guo et al., 2013). By contrast, non-LRF-based

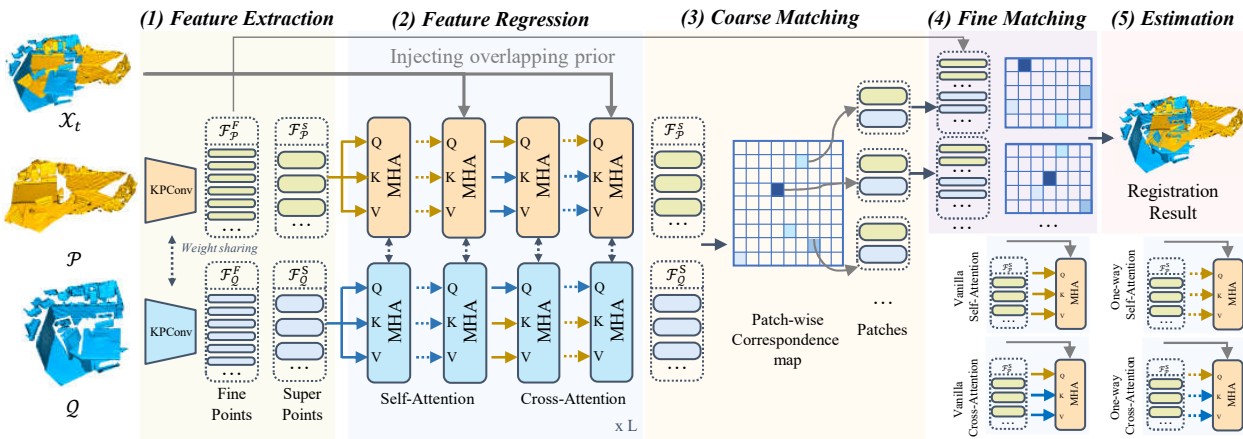

Figure 2: Refinement Network at step index $k$. (1) For two overlapping point clouds $\mathcal{P}$ and $\mathcal{Q}$, we extract super points ($\mathcal{F}_{\mathcal{P}}^{S}$, $\mathcal{F}_{\mathcal{Q}}^{S}$) and fine points ($\mathcal{F}_{\mathcal{P}}^{F}$, $\mathcal{F}_{\mathcal{Q}}^{F}$) using KPConv. (2) We derive matchable superpoint-wise features using Transformers, where MHA stands for multi-head attention. Instead of standard self and cross-attention, we employ our generalized one-way attention, integrating the overlapping information from the previous step $\mathcal{X}_{k-1}$. (3) We compute the patch-wise correspondence map. (4) We propagate patch-wise correspondences to fine correspondences. (5) We obtain the final estimate through a robust estimator. (Best viewed on a screen when zoomed in)

descriptors directly extract features from the raw point cloud. The most notable non-LRF-based methods are histogram-based, computing shape (Frome et al., 2004) or geometry (Rusu et al., 2009; Wu et al., 2017) histograms, and integrating these histograms from key points and their neighbors to form descriptors.

**Learning-based PCR methods** utilize the powerful data-driven ability of deep neural networks for PCR. Several learning-based methods (Choy et al., 2019; Ao et al., 2021) design local extractors to enhance the feature representations. The recent integration of Transformer architectures (Vaswani et al., 2017) has inspired methods (Huang et al., 2021; Yu et al., 2021; Qin et al., 2022; Yang et al., 2022; Chen et al., 2023a; Yu et al., 2023a; Ao et al., 2023) that incorporate attention mechanisms within PCR networks for robust results. In addition, some methods (Choy et al., 2020; Bai et al., 2021; Chen et al., 2022a; 2023b; Zhang et al., 2023) introduce robust estimators to improve the alignment from feature correspondences. Other studies propose deep learning-based end-to-end registration frameworks (Aoki et al., 2019; Wang & Solomon, 2019; Yew & Lee, 2020; Yuan et al., 2020; Yew & Lee, 2022; Chen et al., 2022b), which streamline the registration process. However, these approaches often struggle in challenging scenarios, such as low-overlapping point clouds. Our work is a multi-step registration pipeline, augmenting registration with auxiliary information of estimated overlapping regions. Furthermore, we tailor the registration models to each refinement step, significantly improving the recall rate and robustness.

## 3 Method

Point cloud registration is defined as the process of aligning two partially overlapping point clouds. These are denoted as the source point cloud ($\mathcal{P} = \{p_i \in \mathbb{R}^3 | i = 1, ..., N\}$) and the target point cloud ($\mathcal{Q} = \{q_j \in \mathbb{R}^3 | j = 1, ..., M\}$). The goal is to accurately recover the ground-truth relative rigid transformation, $\mathcal{X}^* = \{R, t\}$, which aligns their overlapping region.

We propose a novel multi-step registration framework, $\mathcal{F} = \{f_k(\mathcal{X}, \cdot) | k = 1, ..., K\}$, to adaptively refine the registration as

$$\mathcal{X}_k = f_k(\mathcal{X}_{k-1}, \mathcal{P}, \mathcal{Q}), \tag{1}$$

where each step $f_k$ utilizes the transformation estimate from the preceding step as auxiliary information. It is important to note that when all the steps are identical ($f_1 = f_2 = \ldots = f_K = f$), the framework $\mathcal{F}$ simplifies to a repetitive iterative method, akin to PEAL (Yu et al., 2023b).

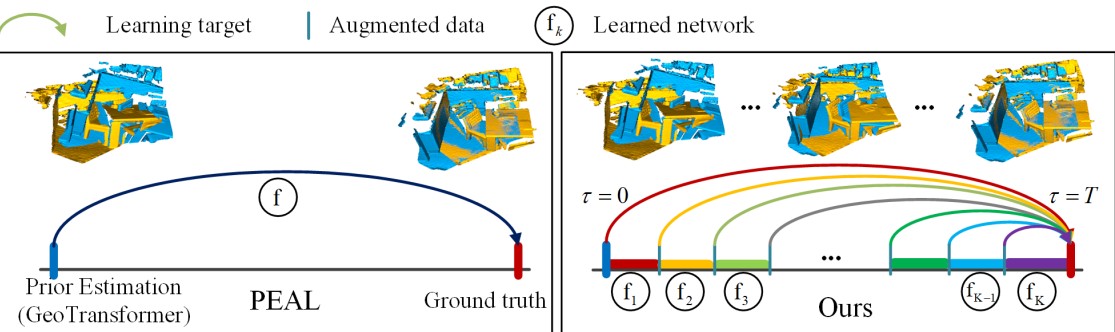

Figure 3: 1D toy example. PEAL trains the network to map prior to ground truth. We propose to sample new prior transformations by interpolating between the GeoTransformer estimate and the ground truth. Upper row: Illustration of the point cloud alignment transitioning from the GeoTransformer estimate to the ground truth. (Best viewed on a screen when zoomed in)

The proposed framework for point cloud registration consists of three parts. 1) Prior (input alignment) creation (Section 3.1), which uses a degradation function, denoted as $\mathcal{E} = \{\epsilon(\mathcal{X}^*, \tau)\}$, to generate a series of rigid transformations with varying levels of accuracy. Here, $\tau \in [1, ..., T]$ is the accuracy level, that higher indicates more accurate prior, and $T$ is the number of the accuracy levels. 2) Training (Section 3.2), which employs the obtained rigid transformation, represented by $\tilde{\mathcal{X}}_{k-1} = \epsilon(\mathcal{X}^*, k)$, to train the registration models $\mathcal{F}$ across different step indices. 3) Inference (Section 3.3), which utilizes the adaptive refinement network to obtain the final estimate $\hat{\mathcal{X}}$.

## 3.1 Creating Point Cloud Prior Transformations

As illustrated in Fig. 3, PEAL (Yu et al., 2023b) trains the network $f$ to map prior to ground truth. This, however, has a limitation: The model does not adjust to the improved quality of the prior during the multi-step refinement process. To adapt the multi-step refinement network to each step, one should specify the refinement $f_k$ that each step should perform.

Consequently, we introduce a degradation function $\epsilon(\mathcal{X}^*, \tau)$ that generates point cloud prior transformations based on the accuracy level $\tau$. Notably, the number of accuracy levels $T$ is much larger than the number of models $K$, producing diverse samples. The degradation function should meet two criteria. Firstly, a higher accuracy level should correspond to a more accurate prior transformation, as each step aims to refine the result from the previous step. Secondly, the initial prior should be a coarse alignment, as we do not expect the network to learn to utilize totally random overlapping information.

Taking these requirements into account, we propose to sample priors for each accuracy level $\tau$ by interpolating between the GeoTransformer prior and the ground truth. Given the non-linearity of the transformation space, we define the degradation within the spherical linear space (Yew & Lee, 2022).

Formally, to define the interpolation, we first define a discrete time step range $[0, 1, ..., T]$. Then, we define the interpolation for both rotation and translation with an accuracy level $\tau \in [0, 1, ..., T]$ as

$$
\begin{aligned}
\mathbf{R}_\tau &= Slerp(\mathbf{R}_{\text{quat}}^{\text{prior}}, \mathbf{R}_{\text{quat}}^{\text{gt}}; \alpha_\tau), \\
\mathbf{t}_\tau &= (1 - \alpha_\tau) \cdot \mathbf{t}^{\text{prior}} + \alpha_\tau \cdot \mathbf{t}^{\text{gt}},
\end{aligned}
\tag{2}
$$

where $\tau$ indicates the accuracy level of the generated prior alignment and $\alpha_\tau = \frac{\tau}{T}$; $Slerp(\cdot, \cdot; \cdot)$ is the spherical linear interpolation (Shoemake, 1985); $\mathbf{R}_{\text{quat}}^{\text{gt}}$ and $\mathbf{R}_{\text{quat}}^{\text{prior}}$ are the quaternion representation of the prior and ground-truth rotation; $\mathbf{t}^{\text{prior}}$ and $\mathbf{t}^{\text{gt}}$ are the prior and ground-truth translation. As $\tau$ decreases, $\alpha_\tau$ approaches 0, bringing $\mathbf{R}_\tau, \mathbf{t}_\tau$ closer to the prior rigid transformation. Conversely, a larger $\tau$ shifts $\mathbf{R}_\tau, \mathbf{t}_\tau$ towards the ground truth. The $Slerp$ function facilitates a smooth transition in rotation between these states.

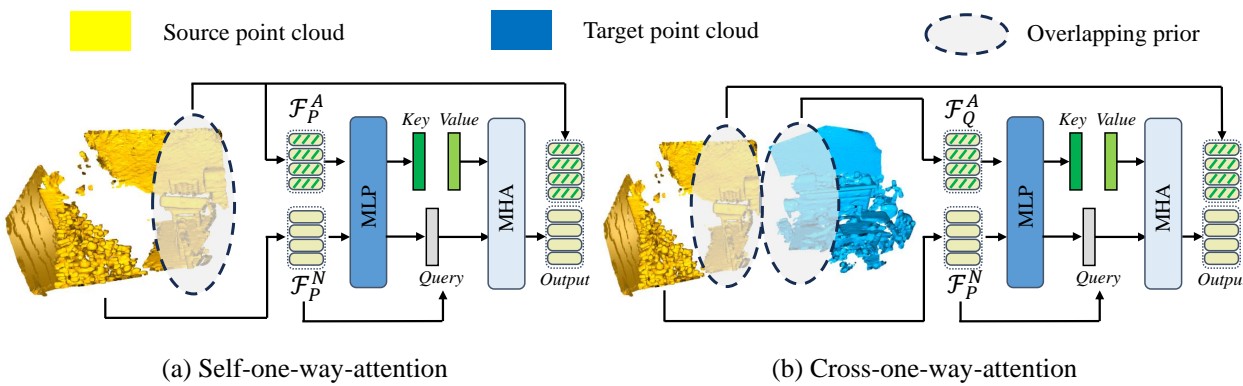

(a) Self-one-way-attention          (b) Cross-one-way-attention

Figure 4: One-way attention (taking the source point cloud $\mathcal{P}$ as an example). (a) The self-one-way-attention uses the feature of anchor points ($\mathcal{F}_{\mathcal{P}}^A$) as Key and Value in the attention operation to compute the feature of non-anchor points, reducing the impact of non-overlapping regions on the learned features ($\mathcal{F}_{\mathcal{P}}^N$). (b) The proposed cross-one-way-attention utilizes the anchor point features in the target point cloud ($\mathcal{F}_{\mathcal{Q}}^A$) as Key and Value, further considering interactions between the two point clouds in the one-way-attention.

## 3.2 Adaptive Multi-step Refinement

**Network backbone.** We adopt GeoTransformer (Qin et al., 2022) as the backbone of the refinement process, illustrated in Fig. 2. The network comprises five stages which we now describe. (1) Feature extraction employs KP-Conv (Thomas et al., 2019), a point-based convolutional network to extract features. This stage gradually down-samples the point cloud in its encoding part to extract sparse points, called superpoints ($\mathcal{P}^S$, $\mathcal{Q}^S$), and features ($\mathcal{F}_{\mathcal{P}}^S$, $\mathcal{F}_{\mathcal{Q}}^S$). It then contains a decoding part, which up-samples the superpoints to yield the final fine points ($\mathcal{P}^F$, $\mathcal{Q}^F$) and features ($\mathcal{F}_{\mathcal{P}}^F$, $\mathcal{F}_{\mathcal{Q}}^F$). Using the point-to-node strategy (Li et al., 2018), each fine point is associated with a superpoint, and each superpoint forms a point patch. (2) Feature interaction leverages multi-head attention operations on the two sets of superpoints to learn matchable features, with $L$ interleaved self and cross-attention layers (see Fig. 2). The self-attention and cross-attention module contain both vanilla attention and the proposed one-way attention (Fig. 4). (3) Coarse matching determines the correspondence map through feature similarity, matching the superpoints by finding the top-k entries. (4) Fine matching refines this alignment at the point level within the patch correspondences established by coarse matching, resulting in precise correspondences. (5) Rigid transformation estimation utilizes the fine correspondences with a robust estimator, such as RANSAC (Fischler & Bolles, 1981) or LGR (Qin et al., 2022), to estimate the rigid transformation.

**Registration process.** After generating the prior data in $T$ different levels of accuracy, we linearly divide this data into $K$ ($K < T$) groups (see Fig. 3). Each group will be utilized to train a specific refinement network, while all models sharing a GeoTransformer backbone. For each accuracy level, the model index is $k = \lceil (\tau/T) * K \rceil$, where $\lceil \cdot \rceil$ denotes the ceiling function.

Moreover, unlike the standard GeoTransformer, which processes only two point clouds, our refinement network integrates the previous estimates. More specifically, we incorporate overlapping information through a one-way attention mechanism, akin to the masking technique prevalent in Transformer models (Cheng et al., 2022). One-way attention gives higher weights to specific regions determined by priors. PEAL (Yu et al., 2023b) introduces one-way attention to PCR and augments the features by focusing on the overlapping regions in the point cloud itself. Inspired by this, we propose a more general one-way attention that attends to both intra and inter-point cloud overlapping regions, arguing that the two overlapping point sets could be complementary, as visualized in Fig. 4. This allows our model to implicitly take the registration result of the last step into account.

To be specific, the input of the one-way attention is the feature of the points in the middle of the network, and one-way attention attempts to establish interactions between point features to update the features. To formulate the one-way attention, given a transformation estimate $\mathcal{X}_\tau = \{R_\tau, t_\tau\}$, we determine the

overlapping points in source point cloud $\mathcal{P}$ as

$$\mathcal{P}_A^S = \{\mathcal{P}_i^S | \max(O(\mathcal{X}_t(\mathcal{P}_i^S), \mathcal{Q}_j^S)) > 0\}, \mathcal{Q}_j^S \in \mathcal{Q}^S, \tag{3}$$

where $\mathcal{X}_t(\mathcal{P}_i^S)$ transforms the patches corresponding to $\mathcal{P}_i^S$ by $\mathcal{X}_t$, and $O(\cdot, \cdot)$ computes the overlap between two point patches. We call these points anchor points. The anchor points in the target point cloud $\mathcal{Q}_A^S$ are defined similarly. We name the remaining points non-anchor points, denoted as $\mathcal{P}_N^S$ and $\mathcal{Q}_N^S$.

Then, we perform one-way attention (OA) between the anchor points and the non-anchor points as shown in Fig. 4. Formally, it can be written as

$$\begin{aligned}
\mathrm{OA}(\mathcal{F}^A, \mathcal{F}^N) &= \mathcal{F}^N + \mathrm{MLP}(F^N + w \times Value^A), \\
w &= softmax(Query^N(Key^A)^T/\sqrt{D}),
\end{aligned} \tag{4}$$

where $\mathcal{F}^A$ and $\mathcal{F}^N$ are the features of the anchor points and the non-anchor points, respectively. Here, we omit the source point and target point subscripts for brevity. $Query^N$, $Key^A$, and $Value^A$ are produced by applying linear transformations to $\mathcal{F}^A$ and $\mathcal{F}^N$. Instead of taking the anchor points and non-anchor points from the same point cloud, as in one-way self-attention, we propose to take the anchor points and non-anchor points from the counterpart point cloud, resulting in one-way cross-attention (see Fig. 4).

### 3.3 Inference

After obtaining the $K$ models trained on the data groups with progressively higher accuracy, we use these models for inference. In our method, the initial model's rigid transformation is set to a prior rigid transformation from GeoTransformer (Qin et al., 2022). Importantly, this initial rigid transformation can be any prior or even pure random noise, and we validate the effect of the input in the experiment section and supplementary material. During the multi-step optimization process, the rigid transformation output by the current model serves as prior for the subsequent model. Finally, the result of the last model is considered as the output of our method.

### 3.4 Loss Functions

To train the refinement model, we adopt the same training losses as GeoTransformer (Qin et al., 2022), which consist of an overlap-aware circle loss $\mathcal{L}_{oc}$ and a point matching loss $\mathcal{L}_p$, i.e.,

$$\mathcal{L} = \mathcal{L}_{oc} + \mathcal{L}_p. \tag{5}$$

The overlap-aware circle loss extends the circle loss (Sun et al., 2020), emphasizing the positive samples with high overlap. The point-matching loss is formulated as a negative log-likelihood loss on the fine point correspondences.

## 4 Experiments

### 4.1 Implementation Details

For a fair comparison with the baselines, we adhere to the same implementation details and experimental setup as GeoTransformer (Qin et al., 2023). We preprocess the point clouds by down-sampling them using voxel grids, setting the voxel size to 2.5cm for the 3DMatch/3DLoMatch benchmarks and 30cm for the KITTI benchmark. Given that the KITTI benchmark typically contains more points per point cloud than 3DMatch/3DLoMatch, we follow GeoTransformer's approach by using a 5-stage KP-Conv for KITTI and a 3-stage KP-Conv for 3DMatch/3DLoMatch. For the data degradation process, we set the accuracy levels $T$ to 1000 to create more diverse data, and this data is divided into $K = 5$ groups for training $K$ models (Section 3.2). We train the KITTI model for 80 epochs and the 3DMatch/3DLoMatch models for 20 epochs, taking around 48 hours on a single V100 GPU. Within each refinement network, the attention module is applied three times (i.e., $L = 3$ in Fig. 2).

| | | | 3DMatch | | | | | | | | | | | | | | |
|---|---|---|---|---|---|---|---|---|---|---|---|---|---|---|---|---|---|
| | | | RR (%) ↑ | | | | | IR (%) ↑ | | | | | FMR (%) ↑ | | | | |
| | Method | Reference | 5000 | 2500 | 1000 | 500 | 250 | 5000 | 2500 | 1000 | 500 | 250 | 5000 | 2500 | 1000 | 500 | 250 |
| Descriptor | FCGF | ICCV2019 (Choy et al., 2019) | 85.1 | 84.7 | 83.3 | 81.6 | 71.4 | 56.8 | 54.1 | 48.7 | 42.5 | 34.1 | 97.4 | 97.3 | 97.0 | 96.7 | 96.6 |
| Descriptor | D3Feat | CVPR2020 (Bai et al., 2020) | 81.6 | 84.5 | 83.4 | 82.4 | 77.9 | 39.0 | 38.8 | 40.4 | 41.5 | 41.8 | 95.6 | 95.4 | 94.5 | 94.1 | 93.1 |
| Descriptor | SpinNet | CVPR2021 (Ao et al., 2021) | 88.6 | 86.6 | 85.5 | 83.5 | 70.2 | 47.5 | 44.7 | 39.4 | 33.9 | 27.6 | 97.6 | 97.2 | 96.8 | 95.5 | 94.3 |
| Descriptor | YOHO | ACM MM2022 (Wang et al., 2022) | 90.8 | 90.3 | 89.1 | 88.6 | 84.5 | 64.4 | 60.7 | 55.7 | 46.4 | 41.2 | 98.2 | 97.6 | 97.5 | 97.7 | 96.0 |
| Transformer-based | REGTR | CVPR2022 (Yew & Lee, 2022) | 92.0 | - | - | - | - | - | - | - | - | - | - | - | - | - | - |
| Transformer-based | Predator | CVPR2021 (Huang et al., 2021) | 89.0 | 89.9 | 90.6 | 88.5 | 86.6 | 58.0 | 58.4 | 57.1 | 54.1 | 49.3 | 96.6 | 96.6 | 96.5 | 96.3 | 96.5 |
| Transformer-based | CoFiNet | NeurIPS2021 (Yu et al., 2021) | 89.3 | 88.9 | 88.4 | 87.4 | 87.0 | 49.8 | 51.2 | 51.9 | 52.2 | 52.2 | 98.1 | 98.3 | 98.1 | 98.2 | 98.3 |
| Transformer-based | GeoTransformer | CVPR2022 (Qin et al., 2022) | 92.0 | 91.8 | 91.8 | 91.4 | 91.2 | 71.9 | 75.2 | 76.0 | 82.2 | 85.1 | 97.9 | 97.9 | 97.9 | 97.9 | 97.6 |
| Transformer-based | OIF-Net | NeurIPS2022 (Yang et al., 2022) | 92.4 | 91.9 | 91.8 | 92.1 | 91.2 | 62.3 | 65.2 | 66.8 | 67.1 | 67.5 | 98.1 | 98.1 | 97.9 | 98.4 | 98.4 |
| Transformer-based | RoITr | CVPR2023 (Yu et al., 2023a) | 91.9 | 91.7 | 91.8 | 91.4 | 91.0 | 82.6 | 82.8 | 83.0 | 83.0 | 83.0 | 98.0 | 98.0 | 97.9 | 98.0 | 97.9 |
| Transformer-based | BUFFER | CVPR2023 (Ao et al., 2023) | 92.9 | - | - | - | - | - | - | - | - | - | - | - | - | - | - |
| Transformer-based | PEAL | CVPR2023 (Yu et al., 2023b) | 94.4 | 94.1 | 94.1 | 93.9 | 93.4 | 74.8 | 81.3 | 86.0 | 87.9 | 89.2 | 98.5 | 98.6 | 98.6 | 98.7 | 98.7 |
| Transformer-based | SIRA-PCR | ICCV2023 (Chen et al., 2023a) | 93.6 | 93.9 | 93.9 | 92.7 | 92.4 | 70.8 | 78.3 | 83.7 | 85.9 | 87.4 | 98.2 | 98.4 | 98.4 | 98.5 | 98.5 |
| Transformer-based | DiffPCR | Arxiv2024 (Wu et al., 2023) | 94.2 | - | - | - | - | 55.4 | - | - | - | - | 97.4 | - | - | - | - |
| Transformer-based | **Ours** | | 94.4 | 94.3 | 94.5 | 94.0 | 93.9 | 75.0 | 81.6 | 86.3 | 88.2 | 89.4 | 98.3 | 98.3 | 98.3 | 98.3 | 98.3 |

| | | | 3DLoMatch | | | | | | | | | | | | | | |
|---|---|---|---|---|---|---|---|---|---|---|---|---|---|---|---|---|---|
| | | | RR (%) ↑ | | | | | IR (%) ↑ | | | | | FMR (%) ↑ | | | | |
| | Method | Reference | 5000 | 2500 | 1000 | 500 | 250 | 5000 | 2500 | 1000 | 500 | 250 | 5000 | 2500 | 1000 | 500 | 250 |
| Descriptor | FCGF | ICCV2019 (Choy et al., 2019) | 40.1 | 41.7 | 38.2 | 35.4 | 26.8 | 21.4 | 20.0 | 17.2 | 14.8 | 11.6 | 76.6 | 75.4 | 74.2 | 71.7 | 67.3 |
| Descriptor | D3Feat | CVPR2020 (Bai et al., 2020) | 37.2 | 42.7 | 46.9 | 43.8 | 39.1 | 13.2 | 13.1 | 14.0 | 14.6 | 15.0 | 67.3 | 66.7 | 67.0 | 66.7 | 66.5 |
| Descriptor | SpinNet | CVPR2021 (Ao et al., 2021) | 59.8 | 54.9 | 48.3 | 39.8 | 26.8 | 20.5 | 19.0 | 16.3 | 13.8 | 11.1 | 75.3 | 74.9 | 72.5 | 70.0 | 63.6 |
| Descriptor | YOHO | ACM MM2022 (Wang et al., 2022) | 65.2 | 65.5 | 63.2 | 56.5 | 48.0 | 25.9 | 23.3 | 22.6 | 18.2 | 15.0 | 79.4 | 78.1 | 76.3 | 73.8 | 69.1 |
| Transformer-based | REGTR | CVPR2022 (Yew & Lee, 2022) | 64.8 | - | - | - | - | - | - | - | - | - | - | - | - | - | - |
| Transformer-based | Predator | CVPR2021 (Huang et al., 2021) | 59.8 | 61.2 | 62.4 | 60.8 | 58.1 | 26.7 | 28.1 | 28.3 | 27.5 | 25.8 | 78.6 | 77.4 | 76.3 | 75.7 | 75.3 |
| Transformer-based | CoFiNet | NeueIPS2021 (Yu et al., 2021) | 67.5 | 66.2 | 64.2 | 63.1 | 61.0 | 24.4 | 25.9 | 26.7 | 26.8 | 26.9 | 83.1 | 83.5 | 83.3 | 83.1 | 82.6 |
| Transformer-based | GeoTransformer | CVPR2022 (Qin et al., 2022) | 75.0 | 74.8 | 74.2 | 74.1 | 73.5 | 43.5 | 45.3 | 46.2 | 52.9 | 57.7 | 88.3 | 88.6 | 88.8 | 88.6 | 88.3 |
| Transformer-based | OIF-Net | NeurIPS2022 (Yang et al., 2022) | 76.1 | 75.4 | 75.1 | 74.4 | 73.6 | 27.5 | 30.0 | 31.2 | 32.6 | 33.1 | 84.6 | 85.2 | 85.5 | 86.6 | 87.0 |
| Transformer-based | RoITr | CVPR2023 (Yu et al., 2023a) | 74.7 | 74.8 | 74.8 | 74.2 | 73.6 | 54.3 | 54.6 | 55.1 | 55.2 | 55.3 | 89.6 | 89.6 | 89.5 | 89.4 | 89.3 |
| Transformer-based | BUFFER | CVPR2023 (Ao et al., 2023) | 71.8 | - | - | - | - | - | - | - | - | - | - | - | - | - | - |
| Transformer-based | PEAL | CVPR2023 (Yu et al., 2023b) | 79.2 | 79.0 | 78.8 | 78.5 | 77.9 | 49.1 | 54.1 | 60.5 | 63.6 | 65.0 | 89.1 | 89.2 | 89.0 | 89.0 | 88.8 |
| Transformer-based | SIRA-PCR | ICCV2023 (Chen et al., 2023a) | 73.5 | 73.9 | 73.0 | 73.4 | 71.1 | 43.3 | 49.0 | 55.9 | 58.8 | 60.7 | 88.8 | 89.0 | 88.9 | 88.6 | 87.7 |
| Transformer-based | DiffPCR | Arxiv2024 (Wu et al., 2023) | 73.4 | - | - | - | - | 22.5 | - | - | - | - | 80.6 | - | - | - | - |
| Transformer-based | **Ours** | | 80.0 | 80.4 | 79.2 | 78.8 | 78.8 | 49.7 | 55.4 | 61.8 | 64.5 | 66.2 | 86.3 | 85.9 | 86.0 | 86.1 | 85.9 |

Table 1: Results on indoor datasets. The results of the compared methods are taken from their paper. The best scores are in **red**, the second best in yellow.

## 4.2 Indoor Scenes: 3DMatch & 3DLoMatch

**Datasets.** Following previous research (Yu et al., 2021; Qin et al., 2022; Yang et al., 2022), we evaluate our method using the 3DMatch (Zeng et al., 2017) and 3DLoMatch (Huang et al., 2021) benchmarks. These two datasets are created from 62 RGB-D scenes, with 46 scenes used for training, 8 for validation, and 8 for testing. A key difference is that 3DLoMatch features a lower overlapping ratio (10% – 30%) than 3DMatch, which has an overlapping ratio greater than 30%.

**Metrics.** We adopt the evaluation metrics from Predator's(Huang et al., 2021), reporting Registration Recall (RR), Feature Matching Recall (FMR) and Inlier Ratio (IR) across varying numbers of correspondences. RR measures the percentage of point cloud pairs aligned within a specified RMSE (Root Mean Square Error), i.e., RMSE $< 0.2m$). IR quantifies the ratio of correspondences that fall within a residual threshold under the true transformation, while FMR assesses the percentage of point cloud pairs with an IR exceeding 5%.

**Registration results.** Tab. 1 compares our method with recent deep learning-based baselines, including 4 local descriptors (FCGF, D3Feat, SpinNet, and YOHO), and 10 Transformer-based methods (REGTR, Predator, CoFiNet, GeoTransformer, OIF-Net RoITR, BUFFER, PEAL, SIRA-PCR, and DiffPCR). We adhere to the evaluation protocol of Predator (Huang et al., 2021), sampling 5 different numbers of correspondences (5000, 2500, 1000, 500, 250) for each method and evaluating the results. As REGTR and BUFFER directly output the final rigid transformation, they are excluded from correspondence metrics. PEAL can use either 2D or 3D information as prior. Since the official 3DMatch and 3DLoMatch datasets do not give the 2D information, PEAL first generates the 2D prior. Since the remaining methods do not need

|  | 3DMatch | | | 3DLoMatch | | |
|---|---|---|---|---|---|---|
|  | RR | IR | FMR | RR | IR | FMR |
| GeoTransformer | 92.5 | 70.9 | 98.2 | 74.0 | 43.5 | 87.1 |
| PEAL-3D | 94.2 | 73.3 | 98.5 | 79.0 | 49.0 | 87.6 |
| **Ours-3D** | 94.4 | 73.4 | 98.3 | 80.0 | 49.6 | 87.0 |
| PEAL-2D | 94.3 | 72.4 | 99.0 | 81.2 | 45.0 | 91.7 |
| **Ours-2D** | 95.3 | 73.9 | 98.5 | 81.6 | 50.4 | 87.7 |

Table 2: Registration results with the LGR estimator. The best scores are in red, the second best in yellow.

the 2D information, in Tab. 1, we report the results of PEAL using a 3D prior as input for a fair comparison. We compare PEAL-2D with our method in a separate table (Tab. 2). Note that PEAL and our method both adopt iterative optimization, and we use the same number of steps (set to 5) in our experiments.

In PCR, RR is the most critical metric (Yang et al., 2022; Huang et al., 2021), which directly measures the registration success rate (RMSE smaller than a threshold). As shown in Tab. 1, our method achieves the highest RR across different numbers of correspondences. Our method outperforms all one-pass techniques, which provide an estimate with a single-step network. Compared with the multi-step method PEAL, our method still achieves notably better results. We attribute this improvement to our adaptive refinement design and our one-way-cross-attention mechanism. Notably, our method with five iterations achieves an RR of 80.4%, which surpasses PEAL's performance with ten iterations (78.8%) on 3DLoMatch. Our method also outperforms DiffPCR (Wu et al., 2023), which is a Diffusion-based PCR network conditioned on the matching matrix, especially on the 3DLoMatch dataset.

In terms of IR, we achieve the best performance in most configurations and slightly worse than one baseline, RoITr, with 5000 correspondences. It means our approach can establish better correspondences through multi-step optimization, which is the key foundation for the improvement of the final registration performance, as reflected by RR. Feature Matching Recall (FMR) measures whether correspondences' inlier ratio is above a threshold, indicating the percentage of point clouds that are *likely* to be registered. Notably, FMR does not test if the actual transformation can be determined (Huang et al., 2021). In terms of the FMR, our technique is inferior to some baselines on the 3DMatch/3DLoMatch benchmarks. Given our higher registration success rate, i.e., RR, we speculate that the worse FMR is caused by point cloud pairs where both the baseline methods and ours fail to register, where our method yields lower IR.

GeoTransformer (Qin et al., 2023) proposes an LGR estimator to compute the rigid transformation for coarse-to-fine based methods. We compare the performance of our method with GeoTransformer and PEAL when combined with LGR. Here, we also compare the results of using a 2D prior as input, as done in PEAL. As shown in Tab. 2, our method still yields the highest registration recall on both 3DMatch and 3DLoMatch, no matter whether with a 2D prior or 3D prior.

Additionally, we compare our method with SOTA 3D outlier removal methods, including PointDSC (Bai et al., 2021), SC2-PCR (Chen et al., 2022a) and MAC (Zhang et al., 2023). In contrast to our method, these methods use the rotation and translation error, instead of the RMSE, as the threshold for computing the registration recall. For a fair comparison, we follow the evaluation strategy of MAC to re-compute the registration recall of our method [2], and present the results in Tab. 3. Following the best result they report, the RR of compared methods is obtained by combining it with the GeoTransformer to establish the correspondences. Our method demonstrates substantial improvements over these methods on both the 3DMatch and 3DLoMatch datasets.

Fig. 5 showcases qualitative results of our method applied to point cloud pairs with exceedingly low-overlapping regions. For comparison, we also show the alignment results of our baseline GeoTransformer, CoFiNet, PEAL, and the ground truth. Our method accurately aligns the point clouds in these challenging settings, while GeoTransformer and PEAL get completely wrong results.

---

[2] We use the evaluation protocol in MAC's official code released at https://github.com/zhangxy0517/3D-Registration-with-Maximal-Cliques

|  | 3DMatch | | | | | 3DLoMatch | | | | |
|---|---|---|---|---|---|---|---|---|---|---|
|  | 5000 | 2500 | 1000 | 500 | 250 | 5000 | 2500 | 1000 | 500 | 250 |
| GeoTransformer + PointDSC (Bai et al., 2021) | 95.4 | 95.4 | 95.2 | 95.2 | 94.5 | 77.8 | 77.9 | 77.2 | 76.9 | 75.7 |
| GeoTransformer + SC2-PCR (Chen et al., 2022a) | 95.6 | 95.7 | 95.1 | 95.3 | 94.7 | 78.3 | 78.4 | 77.8 | 77.2 | 76.3 |
| GeoTransformer + MAC (Zhang et al., 2023) | 95.7 | 95.7 | 95.2 | 95.3 | 94.6 | 78.9 | 78.7 | 78.2 | 77.7 | 76.6 |
| **Ours** | 96.9 | 96.9 | 97.0 | 96.6 | 96.5 | 84.4 | 84.4 | 83.8 | 83.0 | 82.5 |

Table 3: Comparison with the recent outlier removal baselines. Registration Recall is reported as the evaluation metric. The best scores are in **red**, the second best in yellow.

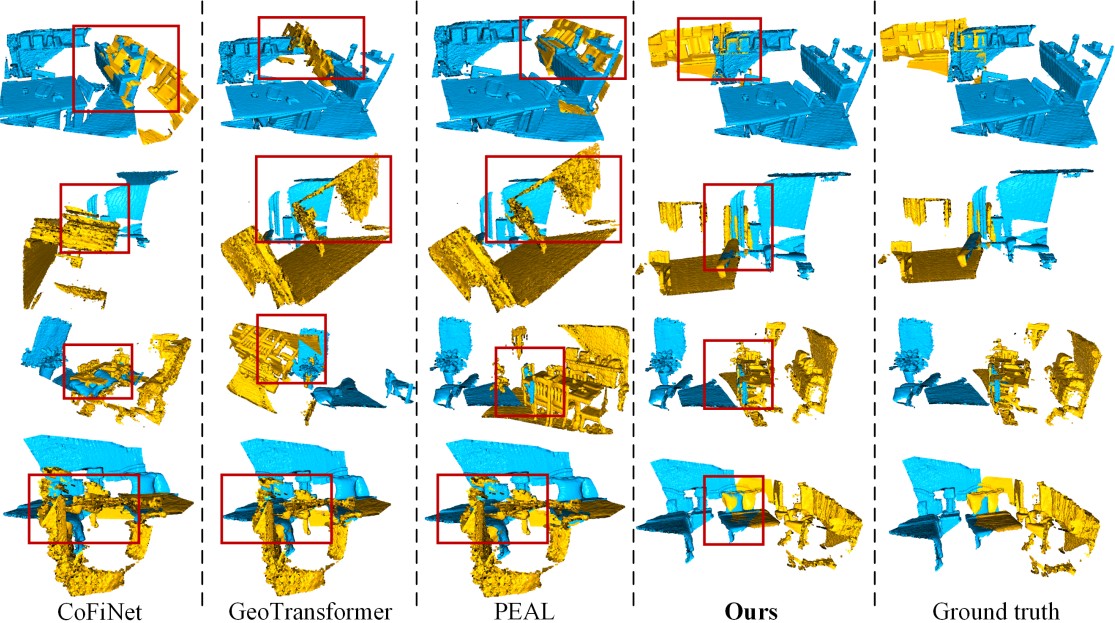

CoFiNet          GeoTransformer          PEAL          **Ours**          Ground truth

Figure 5: Qualitative results of CoFiNet, GeoTransformer, PEAL, and our method compared with the ground-truth alignment. The overlapping areas are highlighted by the red boxes. (Best viewed on a screen when zoomed in)

Despite our method effectively improving registration recall, we present two failure cases in Fig. 6. In both examples, the overlap between the two point clouds is substantially low, and a large portion consists of planes with no distinctive features. Future work on more effective feature extractors may mitigate this issue.

### 4.3 Outdoor Scenes: KITTI Odometry

**Dataset.** KITTI odometry (Geiger et al., 2012) consists of 11 sequences of driving scenes scanned using LiDAR. Following the methodology of previous studies (Qin et al., 2022; Choy et al., 2019), we use sequences 0-5 for training, 6-7 for validation, and 8-10 for testing. We use the optimized ground-truth poses with ICP and use only point cloud pairs that are at least 10m away for evaluation.

**Metrics.** We follow (Huang et al., 2021) and evaluate the methods by three metrics: (1) Relative Rotation Error (RRE), which is the geodesic distance between the estimated and ground-truth rotation matrices; (2) Relative Translation Error (RTE), which measures the Euclidean distance between the estimated and ground-truth translation vectors; (3) Registration Recall (RR), which encodes the fraction of point cloud pairs whose RRE and RTE are both below certain thresholds (RRE$< 5°$ and RTE$< 2m$).

**Registration results.** We compare our network with 9 recent baselines, including FCGF, D3Feat, SpinNet, Predator, CoFiNet, GeoTransformer, OIF-Net, PEAL, and MAC. PEAL does not have open-source code for the KITTI dataset, so we report the result of our implementation. The coarse-to-fine approaches, which include CoFiNet, GeoTransformer, OIF-Net, PEAL, and our method, are compatible with the LGR estimator (Qin et al., 2022). Therefore, we present their results combined with LGR. For the other methods,

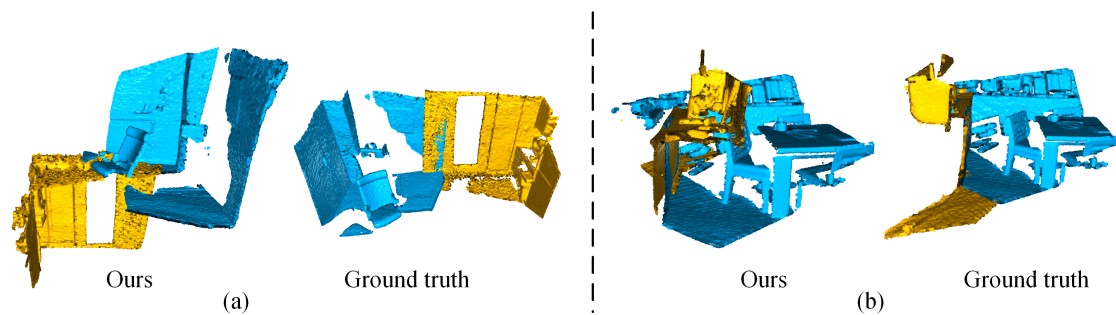

Figure 6: Failure cases of our method. (Best viewed on a screen when zoomed in)

| Methods | Reference | RTE ↓ | RRE ↓ | RR ↑ |
|---------|-----------|-------|-------|------|
| FCGF | ICCV2019 (Choy et al., 2019) | 9.5 | 0.30 | 96.6 |
| D3Feat | CVPR2020 (Bai et al., 2020) | 7.2 | 0.30 | **99.8** |
| SpinNet | CVPR2021 (Ao et al., 2021) | 9.9 | 0.47 | 99.1 |
| Predator | CVPR2021 (Huang et al., 2021) | 6.8 | 0.27 | **99.8** |
| CoFiNet | NeurIPS2021 (Yu et al., 2021) | 8.5 | 0.41 | **99.8** |
| GeoTrans | CVPR2022 (Qin et al., 2022) | 6.8 | 0.24 | **99.8** |
| OIF-Net | NeurIPS2022 (Yang et al., 2022) | 6.5 | **0.23** | **99.8** |
| PEAL | CPPR2023 (Yu et al., 2023b) | 6.8 | **0.23** | **99.8** |
| MAC | CVPR2023 (Zhang et al., 2023) | 8.5 | 0.40 | 99.5 |
| **Ours** | | **6.3** | **0.23** | **99.8** |

Table 4: Results on KITTI odometry. The best scores are in **red**, the second best in yellow.

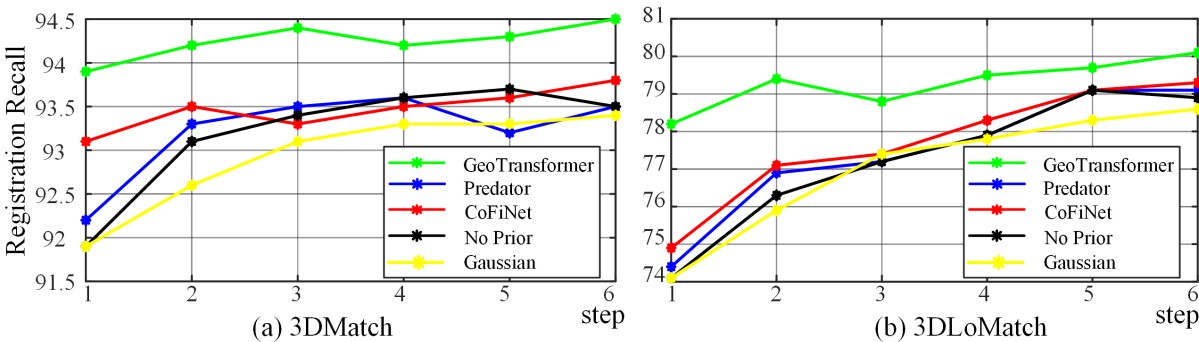

Figure 7: Iteration results using different priors than the training one.

we use RANSAC as their post-processing step. For PEAL and our method, we just perform two steps of iterative optimization because the KITTI dataset is relatively simple. As can be seen in Tab. 4, our method yields a 99.8% RR, which equals the best performance of recent methods. Furthermore, our method achieves the best RRE and RTE.

## 4.4 Ablation Studies

In Tab. 6, we analyze our design choices by ablating different components in our framework. All the experiments use LGR as the post-processing method.

| Overlap (%) / RR↑ | < 15 | 15 - 20 | 20 - 25 | 25 - 30 | > 30 |
|-------------------|------|---------|---------|---------|------|
| GeoTransformer | 57.6 | 72.9 | 78.2 | 84.2 | 92.3 |
| PEAL | 61.9 | 80.4 | 87.6 | 86.2 | 95.7 |
| Ours | **69.4** | **83.1** | **88.7** | **87.8** | **96.5** |

Table 5: Regitration recall in the scenes of different overlap rates. The best scores are in **red**, the second best in yellow.

| | 3DMatch | | | 3DLoMatch | | |
|---|---|---|---|---|---|---|
| | RR | IR | FMR | RR | IR | FMR |
| Full setting | **94.4** | **73.4** | 98.3 | **80.0** | **49.6** | 87.0 |
| w/o cross | 94.3 | 72.6 | 98.3 | 79.5 | 48.5 | 86.7 |
| w/o self | 93.9 | 72.5 | 98.2 | 79.4 | 48.7 | 86.6 |
| w/o self, cross (baseline) | 92.5 | 70.9 | 98.2 | 74.0 | 43.5 | 87.1 |
| Only prior (baseline) | 93.9 | 70.9 | 98.6 | 78.7 | 47.4 | 86.6 |
| Add noise | 92.7 | 67.6 | 97.3 | 76.9 | 44.5 | 85.8 |
| w/o Slerp | 94.1 | 73.0 | 98.2 | 79.6 | 49.2 | 86.5 |

Table 6: Ablations on network components and degradation schemes. **Full setting:** The final version of our method. **Only prior**: Directly using the prior as input without degradation scheme. **Add noise**: Using Gaussian noise as degradation function. The best scores are in red, second best in yellow.

**Evaluation in different overlap rates.** An important aspect of evaluating a network is to measure whether it is robust to noise. For the point cloud registration task, the overlap rate can reflect the noise level. Thus, in Tab. 5, we provide a detailed evaluation by dividing 3DLoMatch into different groups by overlapping ratio, similarly to (Chen et al., 2022a). As reported in the table, our method consistently outperforms PEAL and GeoTransformer, demonstrating our method's robustness, particularly in scenarios with smaller overlaps.

**Initialization with different pretrained models.** In our method, we use the results obtained by the pretrained GeoTransformer as input. To illustrate generalization ability, we test other networks to generate the prior for our method, including Predator (Huang et al., 2021) and CoFiNet (Yu et al., 2021). Furthermore, we also design two control experiments: 1) providing only an identity matrix, not an actual prior; 2) randomly sampling a rigid transformation from a Gaussian distribution as prior. In Fig. 7, we report the results at different iterations. Since Predator and CoFiNet perform worse than GeoTransformer, the performance of using their results as prior is close to using the identity matrix. Through multi-step optimization, even without prior, our method yields near 93.5%/79.0% RR on 3DMatch and 3DLoMatch, respectively, which is much better than the results of GeoTransformer (92.0%/75.0%). This showcases our method's strong generalization to different types of priors.

**Design of the refinement network.** Our method's refinement network for each timestep leverages both one-way-self-attention and our novel one-way-cross-attention. To analyze the effectiveness of these two modules, we remove them separately and summarize the results in Tab. 6. The results indicate that integrating either module enhances the overall performance, with the best results achieved when both are employed.

**Degradation schemes.** In our method, we propose a novel degradation scheme for rigid transformation. Different from the commonly used strategy in generation tasks that relies on random noise as the initial step for training, we interpolate between prior information and ground truth at different time steps as the noisy training data for the refinement network. To validate the importance of the degradation scheme, we also use two different degradation schemes to generate the training inputs. The baseline consists of directly using the prior as the input, akin to PEAL, and a naive strategy is to add noise to the ground truth. As shown in Tab. 6, simply adding noise as degradation scheme yields even worse results than the baseline. The proposed deterministic degradation scheme considers the distribution of the results, and (Full) leads to the highest results in all three metrics on both the 3DMatch and 3DLoMatch benchmarks.

Finally, we remove the Slerp interpolation function and directly use the Euler angle as the interpolation function, and the performance also decreases, because the Slerp function can ensure the linear during the data degradation process.

## 5 Discussions

**Efficiency overhead.** Similar to our baseline PEAL Yu et al. (2023b), multi-step methods are generally slower than single-step methods. As shown in Supp-Tab. 8, our method is consistently more efficient than PEAL with equivalent or better registration results. A promising future direction could be to employ smaller

networks and compression strategies, such as network pruning, to strike a balance between recall rate and speed.

**Storage overhead.** Our pipeline requires storing multiple models and we tested converting the pretrained models from Float32 to Float16, which reduced the storage size by 50% yet maintained an RR↑ of **_80.0%_** on 3DLoMatch, outperforming the PEAL (Yu et al., 2023b) baseline without compression (**_79.0%_**).

# 6 Limitation and Conclusion

Our network exhibits dataset bias, necessitating the training of specific models for each application domain, such as indoor scenes and driving scenes, which is a common issue for learning-based methods.

In summary, we propose an adaptive multi-step refinement network for robust point cloud registration in this paper. We introduced technical innovations of conditioning the refinement network on different steps, and we leveraged a novel training strategy to create point cloud pairs with varying levels of accuracy. Despite being conceptually simple and having minimal overhead, compared to our baseline, our method achieves state-of-the-art registration recall on the 3DMatch/3DLoMatch and KITTI benchmarks, with a notable absolute registration recall improvement of **_1.2%_** on the challenging 3DLoMatch benchmark. This work also demonstrates the potential of the iterative refinement idea from classical point cloud registration methods in learning-based systems.

# 7 Acknowledgements

We appreciate that Dr. Junle Yu and Prof. Wenhui Zhou provide us with the code of PEAL and 2D prior data. Zhi Chen is supported by China Scholarship Council (CSC). This work was supported in part by the Swiss National Science Foundation via the Sinergia grant CRSII5-180359.

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

## A    Overview of this Supplementary

In the following sections, we first provide a comparison with classical non-learning-based PCR algorithms, followed by more specific comparisons with PEAL, and then present more qualitative results.

## B    Comparison with non-Learning-based Methods

| Methods | ICP | Go-ICP | Super4PCS | FGR | Ours |
|---------|-----|--------|-----------|-----|------|
| RR | 6.59 | 22.9 | 21.6 | 42.7 | **94.4** |

Table 7: Comparison of registration recall with classic non-learning methods on the 3DMatch dataset. All values are sourced from the official leaderboard of 3DMatch. The highest scores are highlighted in **bold**.

For interested readers, we present a comparison with classic non-learning-based methods (Tab.7), including ICP (Besl & McKay, 1992), Go-ICP (Yang et al., 2015a), Super4PCS (Mellado et al., 2014), and FGR (Zhou et al., 2016). Our method significantly outperforms non-learning methods because of effective learned features.

## C  Per-step Comparison with PEAL

We present a step-wise registration recall comparison with PEAL. As shown in Fig. 8, our method yields consistently better results. Notably, our method with two steps achieves better results than PEAL with more than 6 steps, highlighting our method's efficacy.

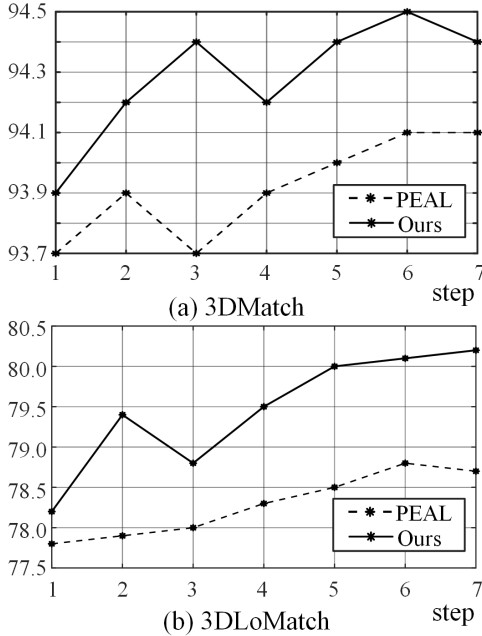

(a) 3DMatch

(b) 3DLoMatch

Figure 8: Multi-step registration recall comparison with PEAL. In both the 3DMatch and 3DLoMatch benchmarks, our method consistently surpasses PEAL. Notably, our method with two steps outperforms PEAL with more than 6 steps.

## D  Computation Time

We present the time cost of our method in Tab. 8, and compare it with two baselines (GeoTransformer and PEAL). As a reference, we also put the registration recall (RR) of the methods[3]. Our method has a higher computational cost than GeoTransformer in 1-step inference, due to additional network components of one-way attentions, while being comparable with PEAL. Note that since our method boosts performance with more steps, we can use the number of steps to run as a parameter for different application scenarios. In addition, our method can get a higher RR with 2-step optimization than PEAL with 5 steps.

| | 3DMatch | | 3DLoMatch | |
| --- | --- | --- | --- | --- |
| | RR | Time (sec.) | RR | Time (sec.) |
| GeoTransformer | 92.0 | **0.296** | 74.0 | **0.284** |
| GeoTransformer + PEAL 1-step | 93.7 | 0.663 | 77.8 | 0.642 |
| GeoTransformer + PEAL 5-step | 94.0 | 2.131 | 78.5 | 2.074 |
| GeoTransformer + Ours 1-step | 93.9 | 0.625 | 78.2 | 0.620 |
| GeoTransformer + Ours 2-step | 94.2 | 0.954 | 79.4 | 0.956 |
| GeoTransformer + Ours 5-step | **94.4** | 1.939 | **80.0** | 1.964 |

Table 8: Comparison of inference time efficiency between GeoTransformer, PEAL and our method. The best scores are in **bold**.

---

[3]For GeoTransformer and PEAL, we use their official code with default settings on the same platform as ours.

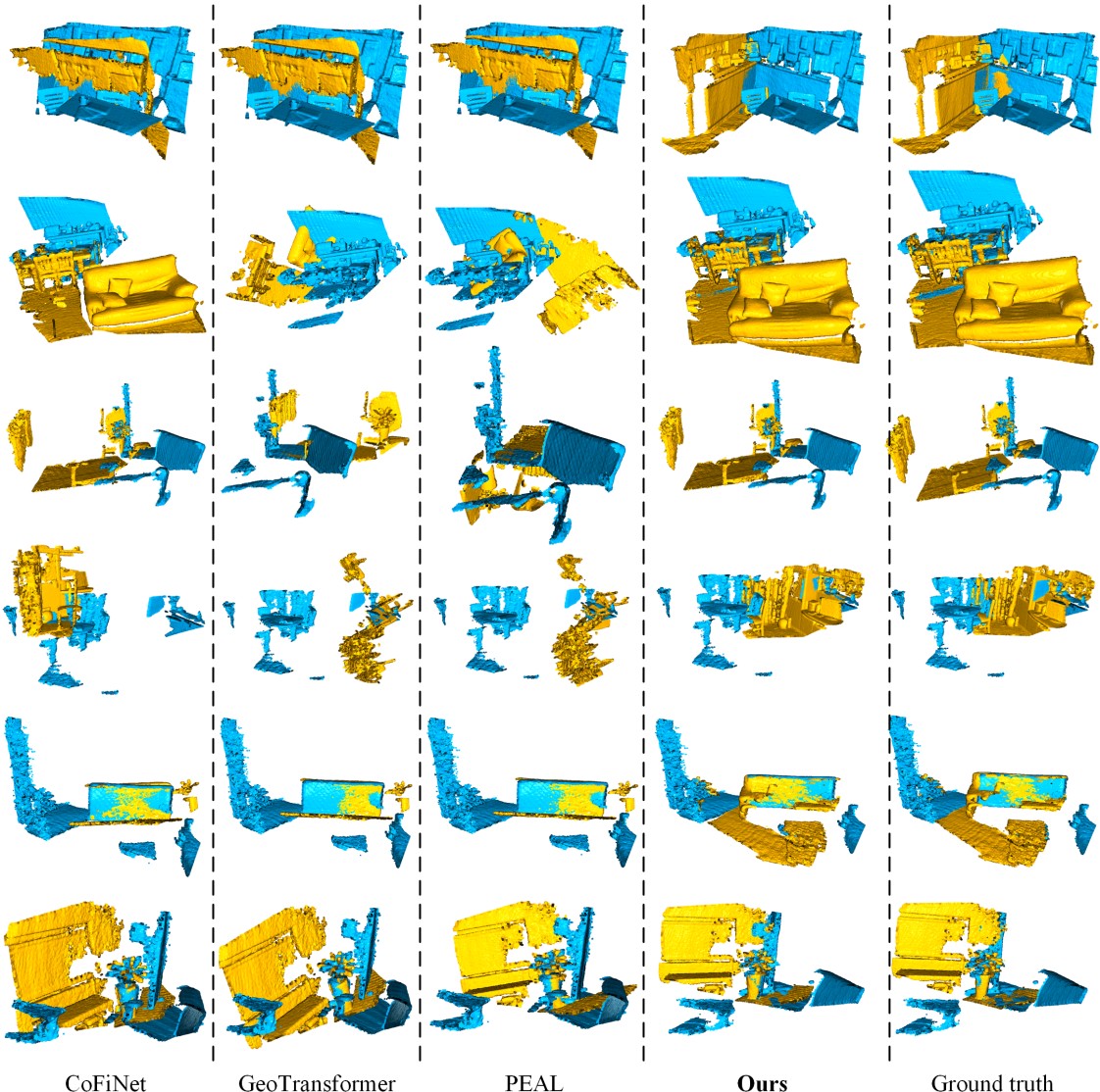

CoFiNet     GeoTransformer     PEAL     **Ours**     Ground truth

Figure 9: Qualitative comparison of the proposed method with recent methods. From left to right are CoFiNet, GeoTransformer, PEAL, ours, and ground truth alignment. Our method best aligns these challenging low-overlapping point cloud pairs. (Best view on a screen when zoomed in)

## E   Qualitative Results

We present more qualitative comparisons with baselines in Fig. 9. Low-overlapping PCR poses a typical challenge, as non-overlapping regions might be mistakenly matched, leading to incorrect alignments. In contrast, our method effectively aligns these point clouds.

