# OpenReview forum: "Adaptive Multi-step Refinement Network for Robust Point Cloud Registration"
_TMLR — Accepted by TMLR_

### Review · Reviewer_MdNU · 2024-09-13

**Summary Of Contributions:**

This paper proposes a new architecture for point-cloud registration. It trains multiple models, with each model responsible only for predicting the relative transformation at different coarse-to-fine levels. To achieve this, it first generates data by interrelating inputs and the ground truth across different levels. Each model is then trained with only a subset of the dataset.

**Audience:**

Yes

**Broader Impact Concerns:**

Not applicable.

**Claims And Evidence:**

Yes

**Requested Changes:**

I’m curious about the different design choices for dividing accuracy levels. In the current setting, the method generates 1,000 different pairs of (input, ground-truth) data by interpolating the raw inputs and ground truth. However, since the raw inputs have varying overlap rates, the generated pairs will also have different overlap rates even if they are in the same accuracy level. For example, if data A has a 0.1 overlap rate and data B has a 0.5 overlap rate, the inputs for the most fine-grained model will be [0.0 - 0.02] for data A and [0.0 - 0.1] for data B. This might be sub-optimal. Have you considered setting accuracy levels according to overlap rates? Specifically, after augmenting the data, you could divide all the data according to their overlap rates and train models for each overlap rate.

Regarding inference time, did you simply apply the five models sequentially? In this case, after applying the first model, the data might be in the optimal input region for model 3 instead of model 2. How do you address this issue?

Sorry if I missed this experiment: Are there any ablation studies on the number of K (e.g., how many models you used for training)?

Figure 6 is quite interesting. I’m surprised that the performance remains good even with random Gaussian or identity inputs. What are the average overlap rates between inputs and ground truth for identity, random Gaussian, and using GeoTransformer?

Table 6 presents important ablation experiments. What is the difference between “only prior” and PEAL? Why does this entry have lower performance compared to PEAL?

What do the failure cases of the proposed method look like?

Minor question: I’m confused about the sentence, “Notably, our method with five iterations achieves an RR of 80.4%, which surpasses PEAL’s performance with ten iterations (78.8%) on 3DLoMatch.” I only found 80.4 in RR2500 and 78.8 in RR1000. Is this a fair comparison?

**Strengths And Weaknesses:**

In general, I believe this paper is polished and well-executed. It presents an interesting idea for iteratively predicting transformations to achieve fine-grained point cloud registration. It demonstrates better performance compared to existing methods on multiple datasets. More importantly, the improvements are more pronounced when the initial overlap is small. The authors also provide a comprehensive analysis of the proposed method, which I find particularly helpful.

Some design choices require clarification or further experimentation. I will detail my questions in the next “Requested Changes” section, but I will briefly summarize them here:
* The method for dividing accuracy levels needs more clarification and exploration.
* More analysis of inference time is needed.
* Ablation experiments with respect to PEAL should be made easier to understand.

---

> ### Author Response · Authors · 2024-10-21
> **Reply to Reviewer MdNU - Part 1**
>
> We greatly appreciate your thoughtful review, your recognition of our polished and well-executed paper, and your appreciation of our comprehensive analysis. We have provided detailed responses to your comments below and will incorporate your suggestions into our revised version.
>
> **Q1**. I’m curious about the different design choices for dividing accuracy levels. In the current setting, the method generates 1,000 different pairs of (input, ground-truth) data by interpolating the raw inputs and ground truth. However, since the raw inputs have varying overlap rates, the generated pairs will also have different overlap rates even if they are in the same accuracy level. For example, if data A has a 0.1 overlap rate and data B has a 0.5 overlap rate, the inputs for the most fine-grained model will be [0.0 - 0.02] for data A and [0.0 - 0.1] for data B. This might be sub-optimal. Have you considered setting accuracy levels according to overlap rates? Specifically, after augmenting the data, you could divide all the data according to their overlap rates and train models for each overlap rate.
>
> **R1**. Thank you for your valuable suggestions. In fact, we experimented with setting accuracy levels based on overlap rates, but this approach resulted in marginally worse performance. We believe this is because our degradation function reported in the paper aligns better with the inference-time scenarios than the approach of dividing data by overlap rates: During the multi-step registration process, the input is initially processed by the first model. Given the variability in overlap rates of the input, the first model must be capable of handling data with varying overlap rates. Training the model on data divided by fixed overlap rates results in models specialized for specific overlap rates, which does not reflect the variability encountered in real-world scenarios.
>
> **Q2**. Regarding inference time, did you simply apply the five models sequentially? In this case, after applying the first model, the data might be in the optimal input region for model 3 instead of model 2. How do you address this issue?
>
> **R2**. Thank you for your insightful question. In practice, we did find some cases where the result of model 1 is better suited for the input region of model 3 rather than model 2. However, empirically, applying model 3 followed by model 2 still yields accurate results. In terms of quality, our model remains competitive, but in terms of efficiency, our original design might introduce some overhead. Following the suggestion of Reviewer s9A7, we added an early exit mechanism, when the aligned correspondences of current estimation are higher than a threshold, we directly stop the subsequent steps. The registration time decreased from **1.96s** to **1.04s**.
>
> **Q3**. Sorry if I missed this experiment: Are there any ablation studies on the number of K (e.g., how many models you used for training)?
>
> **R3**. On page 16, we provide the results of different models of our method and PEAL. As mentioned in Section 4.1, we set the number, K, to 5. We will improve readability in the revised version.
>
> **Q4**. Figure 6 is quite interesting. I’m surprised that the performance remains good even with random Gaussian or identity inputs. What are the average overlap rates between inputs and ground truth for identity, random Gaussian, and using GeoTransformer?
>
> **R4**. Thank you for your interest in this result! We attribute this robustness to our attention module. Our attention module incorporates both one-way self and cross attention, as well as standard self and cross attention. With random Gaussian or identity inputs, the anchor regions may become very small or even empty. In these cases, the standard attention ensures that all points interact, reducing the role of one-way attention. After the first step, the two point clouds become roughly aligned, allowing the overlapping regions to expand. Subsequently, one-way attention refines the alignment.
>
> **Q5**. Table 6 presents important ablation experiments. What is the difference between “only prior” and PEAL? Why does this entry have lower performance compared to PEAL?
>
> **R5**. In our manuscript, the results of PEAL are taken from their paper, while the results in Table 6 were obtained through our reproduction. We noted this difference and compared our reproduced results with the highest result reported by PEAL in our main analysis.

---

> > ### Author Response · Authors · 2024-10-21
> > **Reply to Reviewer MdNU - Part 2**
> >
> > **Q6**. What do the failure cases of the proposed method look like?
> >
> > **R6**. According to your suggestion, we add some failure cases in Fig. 6.
> >
> > **Q7**. Minor question: I’m confused about the sentence, “Notably, our method with five iterations achieves an RR of 80.4%, which surpasses PEAL’s performance with ten iterations (78.8%) on 3DLoMatch.” I only found 80.4 in RR2500 and 78.8 in RR1000. Is this a fair comparison?
> >
> > **R7**. We apologize for the ambiguity. The performance with ten iterations (78.8%) is taken from their paper. This result is not derived from Table 1 (79.2% our reproducing). We would clarify the difference in the revised version and our conclusion still hold that we five iteration model 80.4% outperforms both 79.2% and 78.8%.

---

### Review · Reviewer_s9A7 · 2024-09-21

**Summary Of Contributions:**

First of all, I would like to clarify that I am not an expert in this topic (point cloud registration).

- This paper aims to solve Point Cloud Registration (PCR) by developing a new multi-step refinement network with a modified attention mechanism for the registration process.
- When the overlapping region between the point clouds is small, recent PCR methods struggle due to the generation of ambiguous features, leading to incorrect matching of non-overlapping points.
The authors aim to resolve this issue by introducing a multi-step refinement network, which is a modified version of PEAL, an existing PCR method.
- Empirical results show that training the proposed network outperforms existing baselines, particularly in cases where the overlap between point clouds is small, improving PCR performance.

**Audience:**

Yes

**Broader Impact Concerns:**

I do not think any ethical concerns exist.

**Claims And Evidence:**

No

**Requested Changes:**

- See Weaknesses and Questions above.
- Presentation: The paper would benefit from clearer formulation. The paper could benefit from clearer formulation. For instance, what are the network’s inputs and outputs? What is the output in Figure 4? Is the cross-one-way-attention in Figure 4 part of the overall process in Figure 2?
- (Minor) The notation "K" is used ambiguously, denoting both the number of models and the key vector in the transformer network. This conflict should be clearly revised.

**Strengths And Weaknesses:**

**Strengths**
1. Addressing a practical challenge: The paper effectively tackles a challenge of PCR when the overlapping region is small, a scenario where existing methods often struggle.
2. Experimental studies
   - Comprehensive design: The experimental setup is well-designed, comparing the proposed method against a sufficient number of baseline algorithms. Furthermore, detailed ablation studies validate the effectiveness of each component of the proposed method.
   - Performance: The algorithm achieves state-of-the-art results on benchmark datasets, outperforming recently developed methods.


**Weaknesses and Questions**
1. Limited novelty: The proposed framework seems to be a multi-step variation of PEAL, with the only difference being the use of the target point cloud in attention calculation. Is there any significant novelty compared to PEAL that I might have missed?
2. Explanation about why the proposed algorithm works well: Why does the proposed approach (the multi-step network and the modified attention mechanism in the registration process) outperform existing methods when the overlapping region is small? An intuitive explanation should clarify this connection.
3. Behavior with accurate priors: How does the method perform when the prior transformation from GeoTransformer is nearly identical to the ground-truth? In such cases, dividing the refinement into multiple steps may seem unnecessary.
4. (Question) Are there optimal transport-based methods applicable to PCR? If so, how do they differ intuitively from ICP? If not, why is optimal transport unsuitable for PCR?

---

> ### Author Response · Authors · 2024-10-21
> **Reply to Reviewer s9A7 - Part 1**
>
> We greatly appreciate your thoughtful review and recognition of the practical challenges of this task, as well as your acknowledgment of our paper’s comprehensive experimental design and state-of-the-art results. We have provided detailed responses to your comments below and plan to incorporate your suggestions into our revised version.
>
> **Q1**. Limited novelty: The proposed framework seems to be a multi-step variation of PEAL, with the only difference being the use of the target point cloud in attention calculation. Is there any significant novelty compared to PEAL that I might have missed?
>
> **R1**. We acknowledge your perspective but we respectfully disagree with the characterization of our method as merely a multi-step variation of PEAL. The key novelty of our approach lies in an adaptive multi-step design that achieves state-of-the-art performance (3DLoMatch recall **80.4%**), outperforming more complex algorithms. In contrast, a straightforward multi-step extension of PEAL would suffer from limited adaptability, resulting in suboptimal performance (3DLoMatch recall **79.2%**), as discussed in the experimental section.
>
> Specifically, our contributions include: 1) **A new coarse-to-fine training procedure**. We design a training pipeline that trains different inference models on data with varying levels of accuracy. These models are utilized at different steps of the inference stage, which forms a gradual optimization process for registration. 2) **A new degradation function**. We design a degradation function that interpolates between the prior and ground truth, enabling a smooth transition between accurate and less accurate priors. Adjusting the interpolation ratio allows us to manipulate the quality of the priors. This degradation function enables the creation of diverse training data with varying accuracies. 3) **A generalized one-way attention mechanism**. We propose a generalized version of one-way attention focusing on the overlapping regions in both the source and target point clouds. We discuss the importance of each component in detail in the ablation study.
>
> **Q2**. Explanation about why the proposed algorithm works well: Why does the proposed approach (the multi-step network and the modified attention mechanism in the registration process) outperform existing methods when the overlapping region is small? An intuitive explanation should clarify this connection.
>
> **R2**. Thank you for your insightful question. 1) **Our method leverages previous step prediction to restrict the searching region in a soft way, resulting in more effective use of the network capacity**. Low overlap scenes are very challenging problems in point cloud registration. As shown in Fig. 1, current methods fail in registrating challenging cases in a single step. We attempt to optimize the results of single-step model, and leads to better results by gradual inference. It achieves better results by utilizing the using information of previous steps during the multis-step optimization, and addresses scenarios that current methods cannot handle. 2) **We carefully design the training procedure and degradation function that are tailored for coarse-to-fine registration**.  Compared with existing multi-step methods, such as PEAL, our framework focuses more on the optimization process by introducing a new coarse-to-fine training procedure, a new degradation function, and a generalized one-way attention mechanism. These improvements further enhance performance.
>
> **Q3**. Behavior with accurate priors: How does the method perform when the prior transformation from GeoTransformer is nearly identical to the ground-truth? In such cases, dividing the refinement into multiple steps may seem unnecessary.
>
> **R3**. Thank you for your valuable suggestions. We did not implement a specific procedure for these cases, as our primary focus is on scenarios where GeoTransformer encounters difficulties. In cases where GeoTransformer performs well, our method may be less efficient. **We have gladly incorporated your suggestion by introducing an early exit mechanism**: when the aligned correspondences of current estimation are higher than a threshold, we directly stop the subsequent steps. The registration time decreased from **1.96s** to **1.04s**, with the same performance.

---

> ### Author Response · Authors · 2024-10-21
> **Reply to Reviewer s9A7 - Part 2**
>
> **Q4**. Are there optimal transport-based methods applicable to PCR? If so, how do they differ intuitively from ICP? If not, why is optimal transport unsuitable for PCR?
>
> **R4**. Optimal transport (OT) is indeed widely used in PCR methods, **Following GeoTransformer, our method also uses OT in the fine matching stage**. We learn the cost matrices between points and use the Sinkhorn algorithm to solve the OT problem, generating the final correspondences. Compared with ICP, OT-based PCR methods differ in two main ways: 1) In ICP, the similarity between points is measured by spatial distance, whereas in OT-based PCR methods, similarity is determined by learned features. 2) ICP performs independent matching, considering only the similarity between individual points in the source point cloud and corresponding points in the target pint cloud. In contrast, OT-based methods generate correspondences by considering the global cost of the entire point cloud.
>
> **Suggestions**.
> Presentation: The paper would benefit from clearer formulation. The paper could benefit from clearer formulation. For instance, what are the network’s inputs and outputs? What is the output in Figure 4? Is the cross-one-way-attention in Figure 4 part of the overall process in Figure 2?
> (Minor) The notation "K" is used ambiguously, denoting both the number of models and the key vector in the transformer network. This conflict should be clearly revised?
>
> **Reply**. Thank you very much for your suggestions. According to your suggestions, we revised the paper carefully:
>
> The input and output is are introduced in Section 3.1, *“Point cloud registration is defined as the process of aligning two partially overlapping point clouds. These are denoted as the source point cloud ($\mathcal P = \{p_{i} \in \mathbb{R}^3 | i = 1, ..., N\}$) and the target point cloud ($\mathcal Q = \{q_{j} \in \mathbb{R}^3 | j = 1, ..., M\}$).
> The goal is to accurately recover the ground-truth relative rigid transformation,* $\mathcal X^*=\{R, t\}$, *which aligns their overlapping region.*”
>
> In Section 3.2, we add “*To be specific, the input of the one-way attention is the feature of the points in the middle of the network, and one-way attention attempts to establish interactions between point features to refresh the features*” and “*The self- attention and cross attention contain both vanilla version of attention, and the proposed one-way attention” to show the output in Figure 4 and the composition of the attention in Figure 2*.
>
> The notation of Key vectorion is revised as “Key” instead of “K”.

---

> > ### Comment · Reviewer_s9A7 · 2024-11-26
> >
> > Thank you for your response.
> > I think this detailed explanation will help make the paper more approachable for non-expert readers in this field.

---

### Review · Reviewer_Wm98 · 2024-12-08

**Summary Of Contributions:**

The article proposes an adaptive multi-step refinement method to tackle point cloud registration, especially when the overlapping region between two point clouds is small. It builds on the current multi-step refinement methods, introducing improvements by designing a customized model for each step. Specifically, the authors incorporate the estimated overlapping region from the last step and the step indices, and train the model with varying registration qualities. Experiments are conducted on standard benchmark datasets.

**Audience:**

Yes

**Claims And Evidence:**

Yes

**Requested Changes:**

See weaknesses.

**Strengths And Weaknesses:**

Strengths:
1. Designing a specialized model for each intermediate refinement step is reasonable, as the input registration qualities vary significantly across different steps.
2. The introduction of step indices and the estimated overlapping region from the last steps, along with training multiple models with transformations of varying qualities, allows the model to focus more effectively on the current stage.
3. There are improved results on both indoor and outdoor datasets.

Weaknesses:
1. My primary concern is aligning the multiple registration models trained during the training stage with the different stages during testing. Specifically, during training, varying qualities are defined by formula (2), where the ground truth $\alpha_{\tau}$ is available. However, in the inference stage, we cannot obtain the registration qualities of the current stage, thus it is unclear which registration model to use. I suspect the authors assume that as iteration steps increase, registration qualities also improve, thus progressively adopting models corresponding to higher qualities during testing. However, there are two issues with this approach: (a) the specific registration qualities are still unknown, making it difficult to precisely determine which model would perform better, and (b) qualities may not improve progressively with each step, as shown in Figure 7. Overall, my question is, how do we determine which model to use at each step during testing?
2. During the testing phase, what would be the impact of using the same model across all steps? Assuming there are 6 steps, if the same model is used for each step, either the first or a middle one, instead of using a different model for each step, how much would the results decline? Specific analysis is requested.

---

> ### Author Response · Authors · 2024-12-08
> **Reply to Reviewer Wm98 - Part 1**
>
> We greatly appreciate your thoughtful review and the time and effort you have dedicated to supporting this research community. We are also grateful for your recognition that our design choices are reasonable and that our method achieves state-of-the-art results. Below, we have provided detailed responses to your comments and are happy to offer further clarifications if there are any remaining concerns or additional questions.
>
> **Q1.** My primary concern is aligning the multiple registration models trained during the training stage with the different stages during testing. Specifically, during training, varying qualities are defined by formula (2), where the ground truth is available. However, in the inference stage, we cannot obtain the registration qualities of the current stage, thus it is unclear which registration model to use. I suspect the authors assume that as iteration steps increase, registration qualities also improve, thus progressively adopting models corresponding to higher qualities during testing. However, there are two issues with this approach: (a) the specific registration qualities are still unknown, making it difficult to precisely determine which model would perform better, and (b) qualities may not improve progressively with each step, as shown in Figure 7. Overall, my question is, how do we determine which model to use at each step during testing?
>
> **R1.** Thank you for your valuable question. In practice, a training-inference discrepancy is an inherent challenge in machine learning research, particularly in demanding tasks such as low-overlapping point cloud registration. This discrepancy is reflected in the generally poorer performance observed in low-overlapping point clouds compared to non-low-overlapping ones, with the prior state-of-the-art achieving a test-time recall rate of only 79.2%.
>
> We observed instances where the results of Model 1 were better suited for the input region of Model 3 rather than Model 2 during inference. However, empirically, applying Model 3 followed by Model 2 still produces accurate results and demonstrates a significant improvement compared to simply repeating the same model, as shown in our baseline method PEAL (3DLoMatch recall: **80.4%** vs. **79.2%**). Furthermore, as noted in our response to your second question below, the results from the four settings (using the first, middle, last, and sequential models) highlight that the key to success lies in leveraging progressively more accurate models.
>
> Your insightful question highlights that our method offers a larger design space compared to vanilla PCR methods, which typically rely on a single model. To address your concern and incorporate Reviewer s9A7's suggestion, we have introduced an early exit mechanism into our framework to improve efficiency by skipping unnecessary models. This mechanism determines whether to terminate the registration process early by checking if the aligned correspondence rate of the current estimation exceeds a predefined threshold (e.g., 0.8). If the threshold is met, the process terminates immediately. This modification enhances the flexibility of our framework during inference and avoids redundant model usage. Furthermore, the early exit mechanism reduces the registration time from **1.96** seconds to **1.04** seconds without any decline in registration metrics.
>
> We sincerely appreciate your academic expertise, which has been instrumental in inspiring this key modification to enhance the efficiency of our method.

---

> ### Author Response · Authors · 2024-12-08
> **Reply to Reviewer Wm98 - Part 2**
>
> **Q2.** During the testing phase, what would be the impact of using the same model across all steps? Assuming there are 6 steps, if the same model is used for each step, either the first or a middle one, instead of using a different model for each step, how much would the results decline? Specific analysis is requested.
>
> **R2.** We appreciate your thoughtful comment. To address this question, we conducted additional experiments to evaluate inference performance using different models: (1) the Adaptive Model with Early Exit, (2) the First Model, (3) the Last Model, (4) the Middle Model, and (5) the Single Model (Baseline PEAL). The registration recall rates (using LGR as post-processing) on the 3DLoMatch benchmark are as follows:
>
> | Model                          | RR   | IR | FMR  |
> |--------------------------------|-------|-------|-------|
> | Adaptive Model with Early Exit | **80.0**   | **49.6**  | 87.2 |
> | First Model                    | 79.5  | 49.2  | 87.0  |
> | Last Model                     | 77.2  | 47.6  | 87.1  |
> | Middle Model                   | 78.8  | 48.8  | 86.7  |
> | Single Model (Baseline PEAL)   | 79.0  |  49.0  |**87.6**  |
>
> As shown in the table above, our method significantly outperforms the other configurations in registration recall rate. This finding supports the intuition that employing a sequence of refinement networks with progressively higher accuracy is critical for achieving superior performance. When using a single middle or last model for multiple steps, the results are worse than the baseline. This is because these models are trained on data mixed with ground-truth transformation information, while the input data during inference consists of raw, real data. The latter models require the preceding models to preprocess the data and optimize their outputs before further refinement.
>
> We sincerely thank you for highlighting this important issue, which has allowed us to refine our analysis and has provided valuable insights for future research directions.

---

### Decision · Action_Editor_ENAc · 2025-02-02

**Recommendation:** Accept as is

**Comment:**

The paper works on the classical problem of point cloud registration, addressing the challenging case when the two point clouds overlap little. To this end, the paper proposes an adaptive multi-step refinement procedure where a novel training mechanism (i.e.,  training on transformations with varying registration qualities) and a generalized one-way attention are proposed. Experiments show the efficacy of the proposed technical contributions.

Reviewers initially raised questions mostly about better clarifying the paper, including difference of the proposed method from a a multi-step variation of PEAL, clarification on details of setting about dividing accuracy levels, the unknown issue of specific registration qualities and correspondingly the determination of which models to use during testing stages. The authors give convincing clarifications and present additional experiments including designing an early exit mechanism into the proposed framework. All three reviewers are satisfied with the revision. AE agrees with the reviewers.

**Audience:**

Yes

**Claims And Evidence:**

Yes

---

> ### Author Response · Authors · 2025-02-10
> **Thank you to the Action Editor and Reviewers**
>
> We sincerely appreciate the action editor and reviewers for their constructive feedback and insightful suggestions, which have been invaluable in improving our work!